

# Molecular characterization and genetic diversity of *Jatropha curcas* L. in Costa Rica

Marcela Vásquez-Mayorga[1], Eric J. Fuchs[2], Eduardo J. Hernández[1], Franklin Herrera[3], Jesús Hernández[4], Ileana Moreira[5], Elizabeth Arnáez[5] and Natalia M. Barboza[1,6,7]

[1] Centro de Investigación en Biología Celular y Molecular, Universidad de Costa Rica, San Pedro, San José, Costa Rica
[2] Escuela de Biología, Universidad de Costa Rica, San Pedro, San José, Costa Rica
[3] Estación Experimental Fabio Baudrit Moreno, Universidad de Costa Rica, Alajuela, Costa Rica
[4] Ministerio de Agricultura y Ganadería, San José, Costa Rica
[5] Escuela de Biología, Instituto Tecnológico de Costa Rica, Cartago, Costa Rica
[6] Escuela de Tecnología de Alimentos, Universidad de Costa Rica, San Pedro, San José, Costa Rica
[7] Centro Nacional en Ciencia y Tecnología de Alimentos, Universidad de Costa Rica, San Pedro, San José, Costa Rica

Corresponding author
Natalia M. Barboza,
natalia.barboza@ucr.ac.cr

## ABSTRACT

We estimated the genetic diversity of 50 *Jatropha curcas* samples from the Costa Rican germplasm bank using 18 EST-SSR, one G-SSR and nrDNA-ITS markers. We also evaluated the phylogenetic relationships among samples using nuclear ribosomal ITS markers. Non-toxicity was evaluated using G-SSRs and SCARs markers. A Neighbor-Joining (NJ) tree and a Maximum Likelihood (ML) tree were constructed using SSR markers and ITS sequences, respectively. Heterozygosity was moderate (He = 0.346), but considerable compared to worldwide values for *J. curcas*. The PIC (PIC = 0.274) and inbreeding coefficient ($f = -0.102$) were both low. Clustering was not related to the geographical origin of accessions. International accessions clustered independently of collection sites, suggesting a lack of genetic structure, probably due to the wide distribution of this crop and ample gene flow. Molecular markers identified only one non-toxic accession (JCCR-24) from Mexico. This work is part of a countrywide effort to characterize the genetic diversity of the *Jatropha curcas* germplasm bank in Costa Rica.

## INTRODUCTION

The use of fossil fuels for energy production is being discouraged because of global warming and fluctuating market prices. This situation has motivated research on alternative fuel sources such as biodiesel from corn or palm oil (*To & Grafton, 2015*). *Jatropha curcas* is being explored as a new biofuel crop (*Islam et al., 2013*). It is planted on approximately 1.8 million ha in Indonesia, China, Brazil and Africa, and has the potential to become a biofuel crop in India and other tropical countries (*Carels, 2013*). *J. curcas*, a member of the Euphorbiaceae family, is native to America and has a pantropical distribution. It grows well under

unfavorable climatic and soil conditions, making it an attractive biofuel crop. Average oil content per seed is 40–45% (*Jongschaap et al., 2007*). Biofuel from this species is similar in quality to biofuels derived from conventional crops like canola, linseed and sunflower, and surpasses the quality of biofuels produced from soybean (*Basili & Fontini, 2012*).

The potential of *J. curcas* has not been fully exploited, mainly because of its variable and unpredictable oil yield that limits large-scale cultivation. Genetic improvement may alleviate this problem; however, characterization of the available germplasm is needed for breeding programs to be efficient (*King et al., 2015*; *Mastan et al., 2012*). Over the past decade, *J. curcas* germplasm has been genetically evaluated in India, China, Brazil, Mexico, Costa Rica and Central America (*Avendaño et al., 2015*; *Basha & Sujatha, 2007*; *China Plant BOL Group et al., 2011*; *Montes Osorio et al., 2014*; *Pecina-Quintero et al., 2014*; *Rosado et al., 2010*; *Wen et al., 2010*). Molecular markers such as RAPDs, ISSRs, AFLPs, genomic simple sequence repeats (G-SSR) and expressed sequence tags-SSR (EST-SSR) have all been used to assess the genetic diversity of *J. curcas* collections. These studies have revealed low levels of genetic variability in India and Brazil (*China Plant BOL Group et al., 2011*; *Rosado et al., 2010*; *Sun et al., 2008*; *Yadav et al., 2011*). Numerous authors consider Mexico and Central America to be the center of origin and diversification (*Abdulla et al., 2009*; *Basha et al., 2009*; *Heller, 1996*; *Openshaw, 2000*; *Pamidimarri, Chattopadhyay & Reddy, 2008*; *Pamidimarri & Reddy, 2014*; *Pecina-Quintero et al., 2011*; *Tatikonda et al., 2009*), and high levels of genetic diversity in Guatemala (*Raposo et al., 2014*) and Mexico support this hypothesis (*Ambrosi et al., 2010*; *Ovando-Medina, Adriano-Anaya & Vásquez-Ovando, 2013*; *Ovando-Medina et al., 2011*; *Pamidimarri & Reddy, 2014*; *Pecina-Quintero et al., 2011*). However, these studies have consistently shown a lack of relationship between the geographic proximity of collection sites and the genetic similarity among accessions, because collection sites rarely represent the place of origin of accessions.

Another limitation to large-scale production of *J. curcas* is the possible toxicity of the seed. Both toxic and non-toxic genotypes of *J. curcas* are known (*Insanu et al., 2013*). Varieties with a non-toxic seed cake would be more readily accepted by local farmers because subproducts of the oil extraction process could be utilized for animal feeding (*Makker & Becker, 2015*). Therefore, it is important to evaluate materials stored in germoplasm banks for non-toxicity. Non-toxic genotypes have been described in Mexican accessions (*Vera-Castillo et al., 2014*). As with variation in toxicity, genotypes with variable seed oil content and number of seeds may surface in other Latin American regions and contribute to *J. curcas* breeding programs all over the world.

In the past decade, SSR markers have proven useful for the analysis of genetic diversity. Simple sequence repeats that identify variability in transcribed genomic regions can be found in EST libraries. They may facilitate the identification of functional candidate genes, increase the efficiency of marker-assisted selection and serve as markers for comparative mapping (*Varshney, Graner & Sorrells, 2005*). Nuclear ribosomal-DNA internal transcribed spacers (nrDNA ITS) are sequence-based markers that have been used in phylogenetic studies and to assess genetic diversity at the species level in a wide range of taxonomic groups (*Nieto-Feliner & Roselló, 2007*). ITS markers have been developed for *J. curcas* (*Pecina-Quintero et al., 2011*) and are believed to have greater discriminatory capacity than plastid

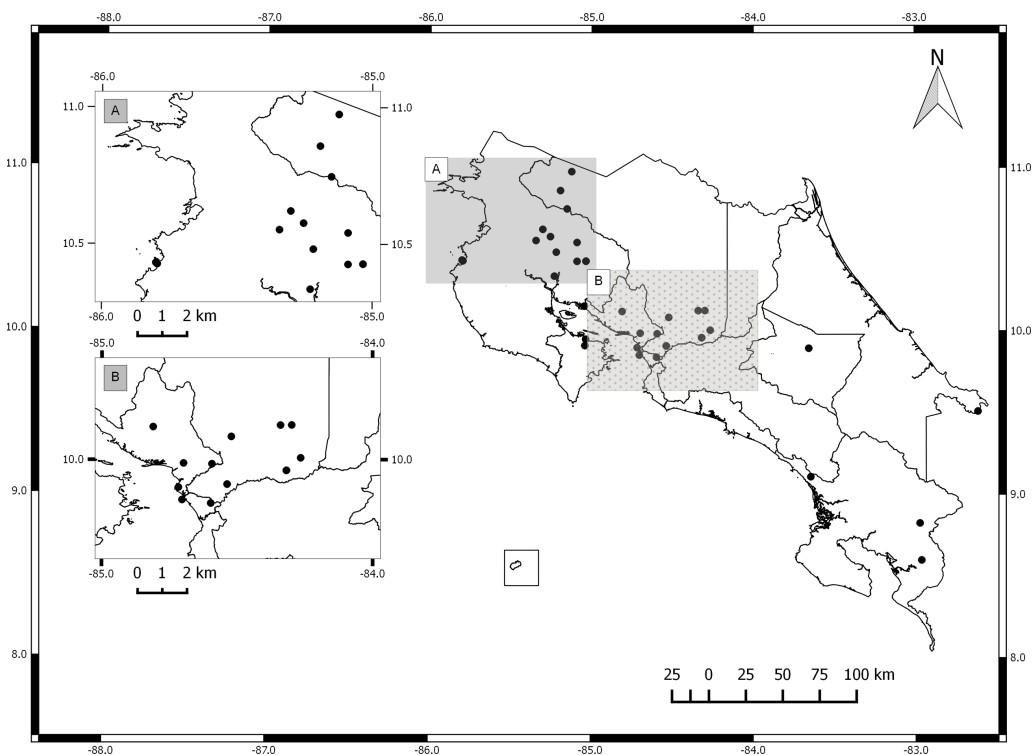

**Figure 1** **Map of Costa Rica with sites where *Jatropha curcas* accessions were collected.** (A) North Pacific Region. (B) Central Pacific Region.

*rbcL* and *matK* markers (*China Plant BOL Group et al., 2011*). SSR and ITS markers used together allow for a better understanding of the evolutionary history of undomesticated species such as *J. curcas.*

Prior to the current study, the genetic variability of the Costa Rican *J. curcas* germplasm bank had not been studied. The present study explores the molecular diversity of accessions using EST-SSR and G-SSR markers and evaluates the phylogenetic relationships between them using nuclear ribosomal ITS markers. Alleles associated with *J. curcas* toxicity were also evaluated with G-SSRs and sequence characterized amplified region (SCAR) markers.

## MATERIALS & METHODS
### DNA collection and extraction
The germplasm bank of *Jatropha curcas* in Costa Rica includes accessions from Honduras, Brazil, India, Mexico, El Salvador, Ecuador, Uganda, Colombia and South Africa. Costa Rican accessions include only spontaneously occurring individuals collected in the field and vegetatively propagated. The collection does not include material from commercial plantations (Fig. 1). International accessions originated from seeds collected from spontaneously occurring plants in each country. All accessions are maintained by vegetative propagation in the germplasm bank located at the Fabio Baudrit Experimental Station (10°00′10.3″N, 84°16′17.6″W) at Universidad de Costa Rica.

For genetic analysis, two young leaves were collected from each of 50 plants representing different accessions in the germplasm bank (Table 1). Vegetative material was frozen and later lyophilized for DNA extraction. Nucleic acids were extracted using the *Möller et al. (1992)* protocol and quantified using a Nanodrop (Thermo Scientific).

## SSR analysis

Eighteen EST-SSRs and one G-SSR marker (*Wen et al., 2010*) were used to analyze the 50 samples (Table 2). PCR was performed in a final volume of 25 µL with 1 × PCR buffer, 400 µM dNTPs, 0.4 µM of each primer, and 1 U Taq DNA polymerase (Thermo Scientific). Amplification conditions included an initial denaturation at 94 °C for 1 min, followed by 35 cycles of 94 °C for 1 min, 42−50 °C for 1 min (depending on the annealing temperature of each primer pair) and 72 °C for 1 min, with a final extension of 72 °C for 10 min. Results were visualized in 4–6% polyacrylamide gels dyed with silver nitrate. To confirm the results, duplicates of 10% of the samples were made. Acrylamide gels were scored manually in the GNU Image Manipulation Program (www.gimp.org) and a data matrix with band size data was created. The SSRs were scored according to amplicon size. The software Microchecker was used to test for null alleles and allelic dropout (*Van Oosterhout et al., 2004*).

## nrDNA-ITS region amplification and sequencing

ITS primers were used to elucidate phylogenetic relationships between accessions. The primer pair JCITS-1-F (5′-ACCTGCGGAAGGATCATTGTCGAAA-3′) and JCITS- 2-R (5′-CCTGGGGTCGCGATGTGAGCGT-3′) was used (*Pamidimarri, Chattopadhyay & Reddy, 2008*) in a PCR reaction with a final volume of 25 µL and a final concentration of 1 × reaction buffer, 1.5 mM MgCl$_2$, 0.2 µM of each primer, 0.2 mM dNTPs, 1 U Taq DNA polymerase (Thermo Scientific) and 50 ng of DNA. The PCR program consisted of an initial denaturation step of 94 °C for 1 min, followed by 30 cycles of 40 s at 94 °C, 65 °C for 1 min and 72 °C for 1 min. The final extension step was 5 min at 72 °C. The PCR products were sent to Macrogen® for sequencing. The obtained sequences were aligned and submitted to Genbank (www.ncbi.com).

## Data analysis

Genetic diversity was quantified as the expected heterozygosity, the Polymorphism Information Content (PIC) and the inbreeding coefficient ($f$). The PIC index describes the probability that two random accessions would have different alleles at a random locus (*Smith et al., 1997*). All estimates were calculated using the Powermarker 3.25 software (*Liu & Muse, 2005*). The validity of all estimates was assessed by means of 50,000 bootstraps.

To determine the genetic relation between accessions, we calculated the $\delta\mu^2$ genetic distance between accessions using the POPULATIONS software (http://bioinformatics.org/~tryphon/populations/). The $\delta\mu^2$ metric developed by *Goldstein et al. (1995)* is based on the stepwise mutation model and estimated distances are a linear function of divergence time; this distance is preferred for taxa that have diverged widely (*Goldstein et al., 1995*). We used the distance matrix to construct a Neighbor-Joining (NJ) tree using the default parameters in POPULATIONS. The standardized genetic distance matrix was also used to perform a Principal Coordinates Analysis (PCA) in GenAlEx 6.5 (*Peakall & Smouse, 2012*).

**Table 1 Germplasm bank information.** Germplasm bank identification number, geographical collection sites and Genbank Accession number of nrDNA-ITS region for each of the *J. curcas* accessions in the Fabio Baudrit Experimental Station used in the study.

| Germplasm code | Country | Location | Genbank accesion number |
| --- | --- | --- | --- |
| JCCR-1 | Costa Rica | Orotina | KU561406 |
| JCCR-3 | Costa Rica | Orotina | KU561407 |
| JCCR-4 | Costa Rica | San Mateo | KU561408 |
| JCCR-5 | Costa Rica | Cañas | KU561375 |
| JCCR-6 | Costa Rica | San Antonio | KU561376 |
| JCCR-7 | Costa Rica | Cañas | KU561377 |
| JCCR-8 | Costa Rica | Abangares | KU561378 |
| JCCR-9 | Costa Rica | Guapinol | KU561379 |
| JCCR-10 | Costa Rica | Cañas | KU561380 |
| JCCR-11 | Costa Rica | San Antonio | KU561381 |
| JCCR-12 | Costa Rica | San Mateo | KU561409 |
| JCCR-13 | Costa Rica | Orotina | KU561382 |
| JCCR-14 | Costa Rica | Turrubares | KU561383 |
| JCCR-15 | Costa Rica | Abangares | KU561384 |
| JCCR-18 | Costa Rica | Bagaces | KU561386 |
| JCCR-19 | Costa Rica | Cañas | KU561387 |
| JCCR-20 | Costa Rica | San Vito | KU561388 |
| JCCR-21 | Costa Rica | Capulín | KU561389 |
| JCCR-23 | Costa Rica | San Antonio | KU561391 |
| JCCR-26 | Costa Rica | Capulín | KU561394 |
| JCCR-28 | Costa Rica | Upala | KU561396 |
| JCCR-29 | Costa Rica | Bagaces | KU561397 |
| JCCR-30 | Costa Rica | Coto 54 | KU561398 |
| JCCR-32 | Costa Rica | Bagaces | KU561400 |
| JCCR-33.1 | Costa Rica | Los Santos | KU561404 |
| JCCR-33.2 | Costa Rica | Los Santos | KU561405 |
| JCCR-34 | Costa Rica | FabioBaudrit | KU561401 |
| JCCR-35 | Costa Rica | Abangares | KU561402 |
| JCCR-36 | Costa Rica | Turrubares | KU561403 |
| JCCR-37 | Costa Rica | Unknown | KU561410 |
| JCCR-40 | Costa Rica | Lagunilla | KU561411 |
| JCCR-41 | Costa Rica | CATIE | KU561412 |
| JCCR-43 | Costa Rica | Turrubares | KU561413 |
| JCCR-45 | Costa Rica | Unknown | KU561415 |
| JCCR-46 | Costa Rica | Diquís | KU561416 |
| JCCR-MIR | Costa Rica | Miramar | KU561422 |
| JCCR-ANA | Costa Rica | Anabel | KU561418 |
| JCCR-MG | Costa Rica | Montaña Grande | KU561421 |
| JCCR-LEP | Costa Rica | Lepanto | KU561420 |
| JCCR-2 | Honduras | Unknown | KU561423 |

**Table 1** (*continued*)

| Germplasm code | Country | Location | Genbank accesion number |
|---|---|---|---|
| JCCR-16 | Brazil | Unknown | KU561385 |
| JCCR-22 | Uganda | Unknown | KU561390 |
| JCCR-24 | Mexico | Unknown | KU561392 |
| JCCR-31 | Mexico | Unknown | KU561399 |
| JCCR-25 | Ecuador | Manabí | KU561393 |
| JCCR-27 | India | Unknown | KU561395 |
| JCCR-INDIA | India | Unknown | KU561419 |
| JCCR-38 | Colombia | Unknown | KU561424 |
| JCCR-44 | El Salvador | Unknown | KU561414 |
| JCCR-47 | South Africa | Unknown | KU561417 |

**Table 2  Primers used for evaluation of *J. curcas*.** EST-SSR and G-SSR primers used for evaluation of *Jatropha curcas* germplasm.

| ID | Forward primer | Reverse primer | $T_A$ (°C)[a] | Expected size (bp) |
|---|---|---|---|---|
| JESR-001 | AACCACAGGAGTTGGTAATG | GAAAGAAGCAACAGAAATGG | 50 | 307 |
| JESR-028 | ACTTCCTTCAGATCATGCAC | CTGGGTAATCTTGTTCCAAA | 52 | 292 |
| JESR-047 | GTTGATACTGGAAGTGAGCC | TGTGTTCAAAGGTGATGAGA | 52 | 398 |
| JESR-086 | TCCCTCTCCTTCAGATTAAA | ATGATAGCCAAACAGCAACT | 54 | 333 |
| JESR-092 | CTCTGAGAATTGAACCATCC | GGGAACAAAGAAATTACTGG | 54 | 378 |
| JESR-093 | CACCTCCCATTAGGGTTT | CTAATCGACGCTGATAATCC | 54 | 239 |
| JESR-095 | AATGAGTCTGACAATCAGGG | GCATGCTCTGTTCTGCTT | 54 | 336 |
| JESR-096 | ACACAAACACAATCAACAGC | CGCGACTCACTTTGTATGTA | 54 | 244 |
| JESR-098 | AGATCACAAGGATCACAAGG | GCAGTTGTCAAACACTAGCA | 54 | 290 |
| JESR-099 | ATAATGGCAAACAAGTGGTC | TGGTAGTGTTGTTCTTGCAG | 54 | 305 |
| JESR-101 | ATCCTAACACAGTTGCCATC | AAACTCAACCAAACCACAAC | 54 | 230 |
| JESR-102 | ATCCTTCTGCAGTAGCCATA | TTATATGCTACACATCAACCTG | 54 | 278 |
| JESR-103 | CAAGTTCGAGGAGTACAAGG | TGTTACAACGAGATGAGTGC | 54 | 292 |
| JESR-104 | CCACAGTTCATCCTCAATTT | GATATTCACTCTGGAACCCA | 54 | 308 |
| JESR-118 | CTAAAGGCTGTGAAGAAGGA | TCCGAGCCAATTTCTTATTA | 54 | 276 |
| JESR-161 | AAGAAGTGTATGGGTTGCAC | TACGATACCTAGGGCTACGA | 56 | 323 |
| JESR-162 | ACTGATGGGTATGTGAGAGG | TTCTTCATCATGGCTACCTT | 56 | 220 |
| JESR-163 | CAGAAACGGAGAGGTCTG | AGATTGGAAGAGGAGAGGAG | 56 | 144 |
| JESR-164 | AGCCCAGTCTCGCGGAAG | CAGTTCCCTTCAGAAGCTC | 56 | 231 |
| JESR-178 | CTTTAGTCCACCTCAAGTGC | TGCAGCAATCAACTCTACTG | 56 | 375 |
| JSSR-203 | ATCCTTGCCTGACATTTTGC | TTCGCAGAGTCCAATTGTTG | 55 | 210 |

**Notes.**
[a] $T_A$, annealing temperature.

For phylogenetic analysis, ITS sequences for *J. curcas* samples from Mexico, India, Cape Verde, Spain, Africa and Madagascar were downloaded from GenBank (http://www.ncbi.nlm.nih.gov). Sequences from Costa Rican samples were edited with BioEdit software version 7.2.5 and aligned using the MAFFT algorithm in the GUIDANCE server. To compare the phylogenetic relationship of GenBank sequences with those from the Costa Rican *J. curcas* germplasm bank, we initially used jModelTest 2.1.7 (*Santorum et al., 2014*) to

**Table 3  Primers used to evaluate toxicity of accessions in the germplasm bank.**

| Primer ID | Primer sequences | $T_m$ (°C) | Expected size non-toxic (bp) | Expected size toxic (bp) |
|---|---|---|---|---|
| JCT27 | F: 5′-CATTAGAATGGACGGCTA-3′<br>R: 5′-GCGTGAAGCTTTGATTTGA-3′ | 60 | 259 | 253 |
| JcSSR-26 | F: 5′-CATACAAAGCCTTGTCC-3′<br>R: 5′-AACAGCATAATACGACTC-3′ | 55 | 210 | 230 |
| JCT31 | F: 5′-TGGAAAACGAATGAGGCTCT-3′<br>R: 5′-GGACACTCTGGAAAGGAACG-3′ | 59 | 214 | 208 |
| ISPJ1 | F: 5′-GAGAGAGAGAGAGAGGTG-3′<br>R: 5′-GAGAGAGAGAGAGAAAACAAT-3′ | 54 | NA[a] | 543 |
| ISPJ2 | F: 5′-GAGAGAGAGAGTTGGGTG-3′<br>R: 5′-AGAGAGAGAGAGCTAGAGAG-3′ | 54 | 1,096 | NA |

**Notes.**
[a] NA, No amplification expected.

define the optimum substitution model for all sequences and the GTR model (*Tavaré, 1986*) with uniform rates was chosen. A maximum likelihood (ML) tree was constructed with 3,000 bootstrap replications and expressed as the number of base substitutions per site using MEGA 6.0 (*Tamura et al., 2013*).

## Toxicity evaluation

Plant toxicity was evaluated by the presence of alleles from three SSR markers (JCT27, JcSSR-26 and JCT31) associated with lack of toxicity (Table 3) (*Vischi, Raranciuc & Baldini, 2013*). PCR amplification was achieved in a final volume of 25 µL with a final concentration of 1 × reaction buffer, 1.5 mM of $MgCl_2$, 0.2 µM of each primer, 0.2 mM dNTPs, 1 U Taq DNA polymerase (Thermo Scientific) and 50 ng/µl DNA sample. The PCR program had an initial denaturation of 94 °C for 1 min, 30 cycles of 40 s at 94 °C, 65 °C for 1 min, 72 °C for 1 min and a final extension of 5 min at 72 °C. Results were visualized by genotyping with fluorescent dyes (FAM, VIC and PET) (*Vischi, Raranciuc & Baldini, 2013*) in a 3130 sequencer (Applied Biosystems). Two SCAR markers (ISPJ1 and ISPJ2) were also used to evaluate alleles for toxicity in all accessions following the protocol of *Basha & Sujatha (2007)*. ISPJ1 amplifies a 543 bp fragment and is specific for toxic genotypes, while ISPJ2 is specific for non-toxic genotypes and amplifies a 1,096 bp fragment. Results were visualized in a 2% agarose gel run for one hour and dyed with GelRed (Biotium). Plants were scored as toxic or non-toxic based on the presence and size of amplicons.

## RESULTS

### Genetic diversity analysis

Genetic diversity was estimated using 18 SSR-ESTs and one G-SSR marker. Average heterozygosity (He) was 0.346 ± 0.062 (±SD). Polymorphism information contents (PIC) ranged from 0.042 to 0.677, with a mean PIC of 0.274 ± 0.165. We did not find evidence of inbreeding $f = -0.102 \pm 0.346$ (Table 4).
**Table 4  Parameters of genetic diversity information obtained.**

| SSR name | He[a] | PIC[a] | $f$[a] |
|---|---|---|---|
| JEST-01 | 0.375 | 0.551 | 0.411 |
| JEST-28 | 0 | 0.043 | 0.001 |
| JEST-47 | 0.050 | 0.048 | −0.0129 |
| JEST-86 | 0 | 0.336 | 0.001 |
| JEST-92 | 0.814 | 0.370 | −0.654 |
| JEST-93 | 0.327 | 0.250 | −0.160 |
| JEST-95 | 0.217 | 0.175 | −0.111 |
| JEST-96 | 0.380 | 0.260 | −0.225 |
| JEST-98 | 0.043 | 0.114 | 0.650 |
| JEST-99 | 0.280 | 0.236 | −0.110 |
| JEST-101 | 0.500 | 0.357 | −0.062 |
| JEST-102 | 0.325 | 0.235 | −0.182 |
| JEST-118 | 0.313 | 0.228 | −0.175 |
| JEST-161 | 0.043 | 0.042 | −0.011 |
| JEST-162 | 0.500 | 0.677 | 0.300 |
| JEST-163 | 0.245 | 0.192 | −0.129 |
| JEST-164 | 0.721 | 0.355 | −0.555 |
| JEST-178 | 0.512 | 0.364 | −0.058 |
| JSSR-203 | 0.933 | 0.375 | −0.862 |
| Mean | 0.346 | 0.274 | −0.102 |
| SD | 0.062 | 0.165 | 0.346 |

**Notes.**
[a]He, Heterozygosity; PIC, Polymorphism information content; $f$, inbreeding coefficient.

The NJ tree did not show a clear clustering pattern (Fig. 2) and clusters did not reflect geographic proximity. Accessions from countries located in close proximity such as Colombia (JCCR-38) and Ecuador (JCCR-25) did not seem to be genetically close to each other. In contrast, samples from distant locations clustered together, for example, India (JCCR-INDIA) and Costa Rica (JCCR-14); South Africa (JCCR-47) and Honduras (JCCR-2); Brazil (JCCR-16) and Costa Rica (JCCR-MIR); and Mexico (JCCR-24) and Ecuador (JCCR-25). Within Costa Rica, there was no evidence of geographic structure. Samples collected from sites separated by more than 300 km grouped together (JCCR-20, JCCR-7).

Our PCA analysis produced comparable results. Genetic information accounted for 41.26% of the observed variance; the first two components explained 19.17% and 11.86% of the total variance, respectively. We did not observe distinct groups of accessions in a biplot of the first two components (Fig. 3). Accessions from the same country did not group together, such as those from Mexico (JCCR-24 and JCCR-31) and India (JCCR-INDIA and JCCR-27 INDIA). Costa Rican accessions were scattered throughout the plot without any discernible pattern. These results were congruent with our NJ tree.

## nrDNA-ITS sequence analysis

As with the NJ and PCA analysis, the observed patterns from the Maximum Likelihood (ML) tree did not reflect the geographic origin of the accessions (Fig. 4) and in most cases, the

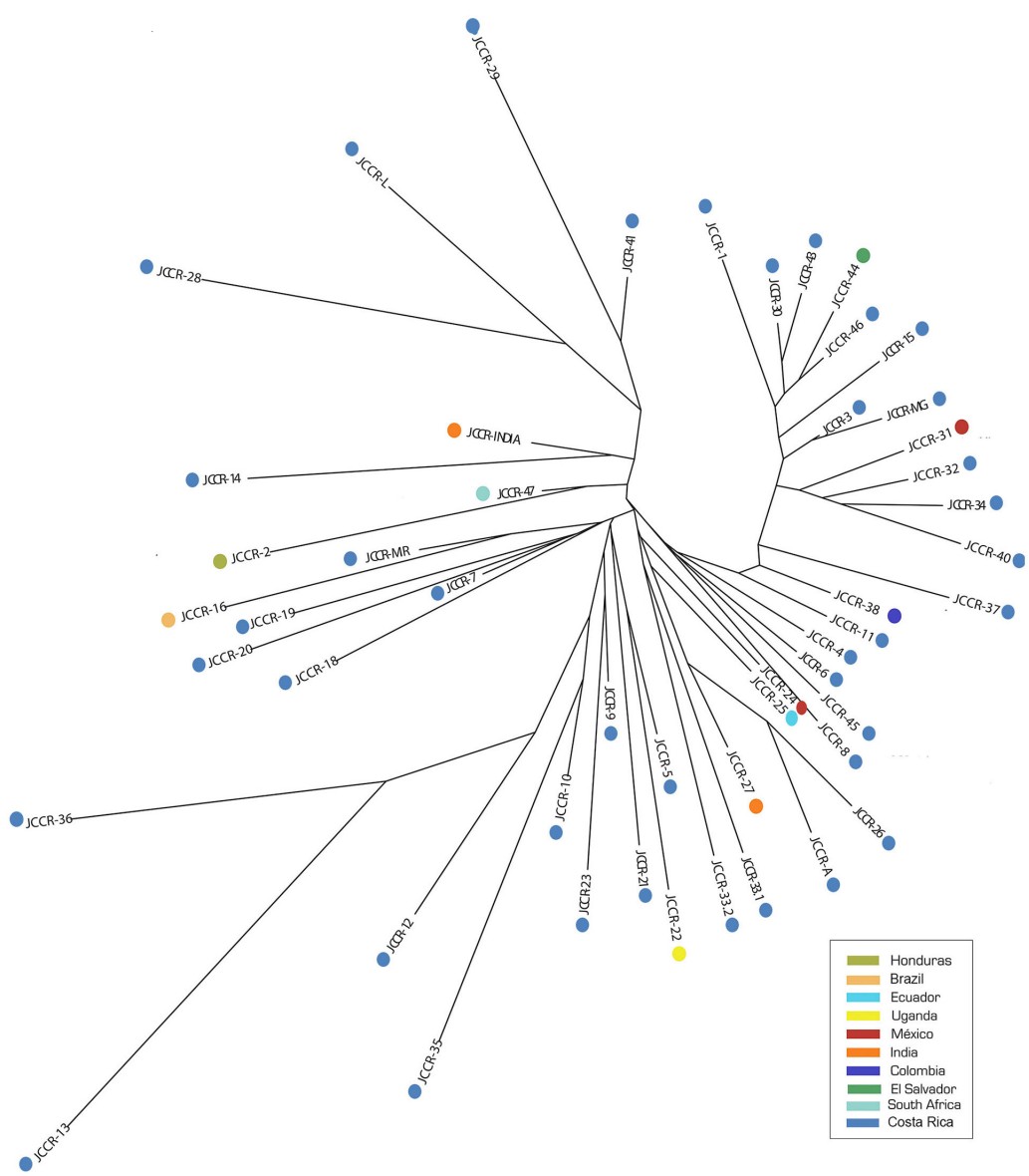

**Figure 2** **Neighbor-Joining (NJ) tree based on a genetic distance matrix from 19 microsatellite data from 50 germplasm accessions of *J. curcas*.**

clades grouped in polytomies. International and Costa Rican accessions from this research were scattered throughout the tree (JCC-38, JCCR-2, JCCRLEP, JCCR33-1, JCCR31, JCCR21, JCCR19, JCCR9, JCCR8, JCCR7, JCCR3), they grouped with sequences from Mexico and with individuals from Spain, Cape Verde, Africa and Madagascar (GenBank accession numbers EU700449, EU700455, EU700446, EU700445, respectively). It is important to mention that the observed clustering of germplasm independent of geographical origin could be an artifact of unequal sampling, as the majority of the samples used in this research were from Costa Rica and only one or two representatives from each of the other countries were included.

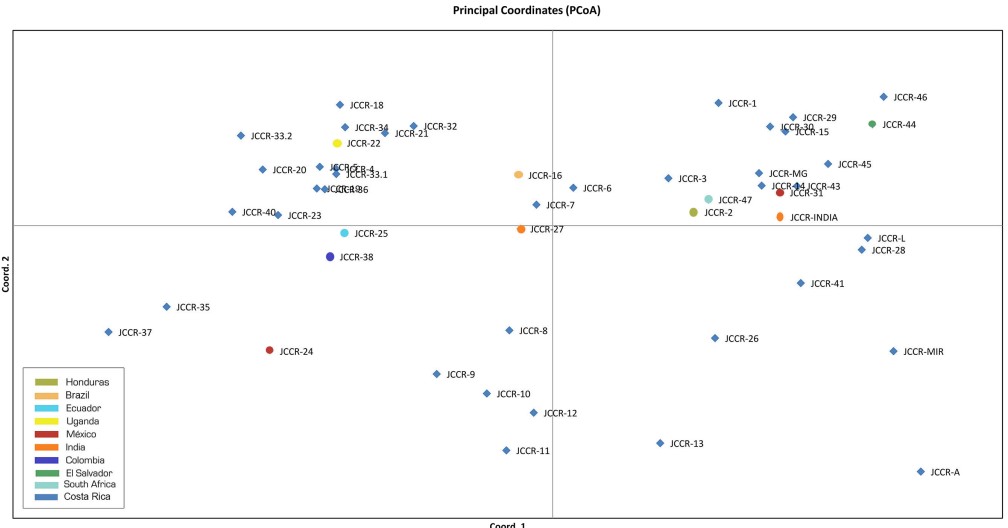

**Figure 3** Principal coordinate analysis created with GenAlEx based on the $\delta\mu^2$ genetic distance estimated in the Populations software with 19 microsatellites from 50 germplasm accessions of *J. curcas* from the Costa Rican germplasm bank evaluated in this study.

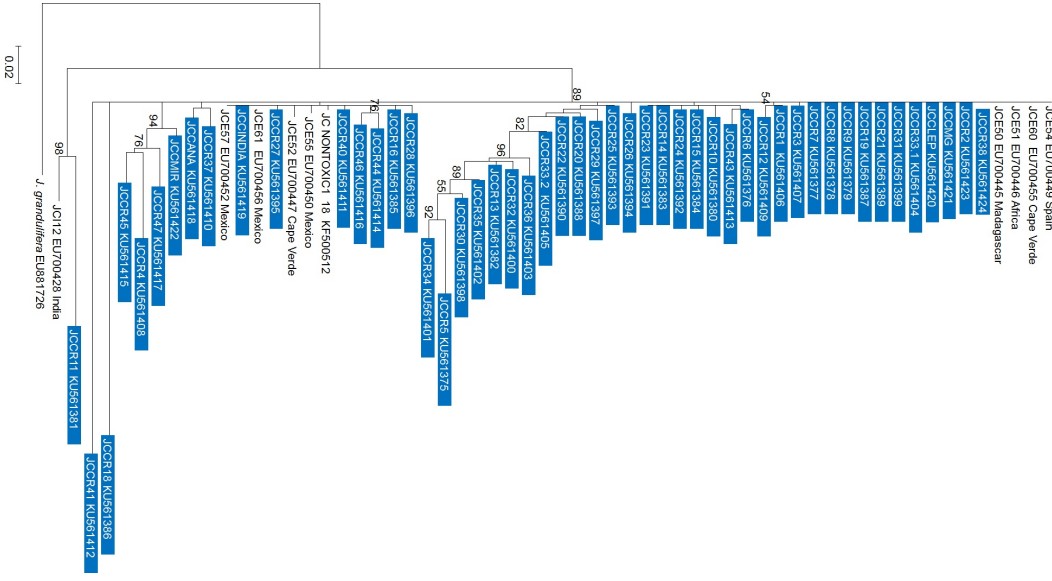

**Figure 4** Maximum likelihood phylogenetic tree generated with 60 *Jatropha curcas* samples. Analyzed sequences represent the nrDNA-ITS region. The tree was constructed with the GTR model using a jModelTest analysis with 3,000 bootstraps and uniform substitution rates in the MEGA 6.0 software. Only bootstrap values higher than 50% are shown. The bar indicates the substitutions per site. Blue boxes show the sequences obtained from the current work in Costa Rica. Other accessions were obtained from Genbank for comparison.

## Toxicity evaluation

A single Mexican accession (JCCR-24) was identified as non-toxic by the SSR primer set (JCT-31, JCT-27 and JcSSR-26) and both SCAR primers ISPJ1 and ISPJ2. ISPJ1 also identified non-toxic genotypes in two other accessions: JCCR-32 and JCCR-43. Using ISPJ2, accessions JCCR-22 and JCCR25 were identified as non-toxic. These results need to be corroborated using *in vivo* assays.

# DISCUSSION

## Genetic diversity of *Jatropha curcas*

Our study is the first to provide genetic diversity estimates for Costa Rican *J. curcas* samples. Based on morphological and molecular evidence, different authors support the idea that Mexico and Central America may be the center of origin and diversification for *J. curcas* (*Pamidimarri & Reddy, 2014*; *Pecina-Quintero et al., 2014*). *Pecina-Quintero et al. (2014)* found high genetic diversity of *J. curcas* in Mexico. *Grativol et al. (2010)* analyzed 332 accessions from 12 locations in Brazil using ISSR primers and reported lower genetic diversity than that reported in Mexico. *Pamidimarri & Reddy (2014)* used RAPD and AFLPs to analyze the molecular diversity of 42 Indian accessions of *J. curcas* and found a mean percentage of polymorphism (PP) of 26.47. In the same study, the average PP of Mexican accessions was 33.18. The mean PP of the germplasm samples excluding the Indian accessions was 35.86, supporting the hypothesis that Indian germplasm is less diverse than germplasm in other regions of the world (*Colombo, Second & Charrier, 2000*; *Ram, Kumar & Bhatt, 2004*). *Montes Osorio et al. (2014)* also found higher genetic diversity in Mexican and Central American accessions compared to those from other parts of the world. Given the observed levels of genetic diversity, Costa Rica may be a secondary center of origin or diversification for this species (*Pamidimarri & Reddy, 2014*; *Pecina-Quintero et al., 2014*). However, comparisons are limited since few studies have evaluated genetic diversity of *J. curcas* germplasm using EST-SSR markers. *Wen et al. (2010)* evaluated 45 accessions from Indonesia, Grenada, South America and two Chinese provinces, and found a mean genetic diversity of 0.3819. The most diverse locations were South America and Yunnan with $H = 0.33$ and $H = 0.3473$, respectively. In another study, 50 EST-SSR markers were used to evaluate 25 Indian accessions and an average He of 0.30 was found (*Yadav et al., 2011*). As in our study, accessions clustered independently of geographic origin. Our PIC estimates (PIC $= 0.274 \pm 0.165$) were comparable to those obtained by *Yadav et al. (2011)* (PIC $= 0.25 \pm 0.16$) and are considered moderately informative (*Botstein et al., 1980*).

In Costa Rica, *Jatropha curcas* is typically not cultivated commercially. Plants usually grow as hedgerows and are occasionally reproduced by farmers through cuttings. Our samples represent plants growing spontaneously in the field; no commercially grown material was included in the study. Therefore, our genetic diversity estimates represent the standing natural variation of this species. However, since EST-SSR markers were developed from expressed sequence tag libraries, they reside within genes and are subject to selection, which reduces unfavorable polymorphisms (*Cova et al., 2012*; *Ellis & Burke, 2007*). EST-SSR markers are less polymorphic than genomic SSRs (*Song et al., 2012*) and consequently, genetic

diversity may have been underestimated in this study. *J. curcas* is a predominantly outcrossing species ($tm = 0.683$) (*Bressan et al., 2013*). As expected, we found no significant evidence of inbreeding ($f = -0.102$). High rates of gene flow should produce low levels of inbreeding, which would result in the low structure suggested by our NJ and ML clustering. Low inbreeding coefficients were also estimated in Mexico and South America (*Ambrosi et al., 2010*). Although inbreeding was negligible in the present study, in other parts of the world such as India and Brazil, lower genetic diversity has been attributed to increased selfing or a high paternity correlation due to the spread of introductions across the country through vegetative propagation, recent common ancestry, drift, and intensive selection of the currently cultivated materials since the time of introduction (*Basha & Sujatha, 2007*; *Bressan et al., 2013*; *Rosado et al., 2010*).

## Phylogenetic analysis of *Jatropha curcas*

Several studies of *J. curcas* have shown that collection sites do not necessarily reflect the genetic origin of accessions (*Ambrosi et al., 2010*; *Maghuly et al., 2015*). Our NJ tree and PCA analysis (Figs. 2 and 3) showed no correlation between genetic similarity and geographic proximity. For example, the two Mexican accessions, JCCR-24 and JCCR-31, clustered in different putative groups. Also, the two Indian accessions (JCCR-27 and JCCR-INDIA) did not seem to be related. *J. curcas* is widely cultivated and plants are exchanged commonly. Accessions from the same country may come from diverse origins and thus may be placed in different clades. Our analysis suggests that collection sites may not necessarily represent local germplasm, but genetically distinct lineages from different geographic regions. Material exchanges between American, African and Asian collections have occurred commonly over the last 200 years (*Heller, 1996*) and may have resulted in founder effects in Africa and Asia (*Henning, 2007*; *Lengkeek, 2007*; *Pamidimarri & Reddy, 2014*). For example, according to *Pamidimarri & Reddy (2014)*, Portuguese seafarers introduced accessions from Mexico and Central America to India through two dispersal routes: one brought *J. curcas* through Africa, Madagascar and finally to India, while the other passed through Spain on its way to India. These migration routes support our findings of a widely dispersed plant with little geographic structure.

The observed ML tree topology may be an artifact of the low level of genetic structure seen in our other analysis. Concurrently, low levels of inbreeding suggest considerable gene flow may be occurring in *J. curcas* in Costa Rica. Although our samples did not cluster similarly across analysis, a general lack of group structure was maintained throughout. The lack of consensus between clustering algorithms may be attributed to the nature of the different markers used. We analyzed multiple EST-SSR loci distributed throughout the genome (*Davies & Bermingham, 2002*; *Pamidimarri & Reddy, 2014*), and we are confident that we have accounted for a significant portion of the genetic variability in this species. Costa Rican diversity estimates may be improved by enriching the germplasm bank with more accessions from the Caribbean and southern parts of the country.

## Toxicity evaluation of *J. curcas*

Our results show that only one of the accessions, Mexican JCCR-24, had all of the alleles that indicate non-toxicity. This accession was previously confirmed as non-toxic by *in vivo*

evaluation (*Basha & Sujatha, 2007*). JCCR-24 could be used as a parental plant in a breeding program to obtain dual-purpose non-toxic plants, thereby increasing the attractiveness of *J. curcas* as a biofuel plant (*King et al., 2013*). In other samples, only one or two toxic alleles were detected, depending on the primers used. This may have been due to variations present in Costa Rican genotypes. SSR markers are very polymorphic (*Powell et al., 1996*). The primers used to detect non-toxicity were developed from Mexican, Asian and African accessions (*Basha & Sujatha, 2007*; *Phumichai et al., 2011*) and their ability to detect toxicity may differ for genotypes from other parts of the world. It is possible that other Costa Rican accessions have alleles that have not yet been identified as indicative of non-toxicity, and in this case, the non-toxic nature of the accessions would have been overlooked. *In vivo* evaluations are needed to confirm this hypothesis.

## ACKNOWLEDGEMENTS

We thank the Estación Experimental Fabio Baudrit Moreno (EEFBM) of the Universidad de Costa Rica for the collection and maintenance of the germplasm bank.

### Funding
This work was supported by FEES-CONARE (736-B1-660). The funders had no role in study design, data collection and analysis, decision to publish, or preparation of the manuscript.

### Grant Disclosures
The following grant information was disclosed by the authors:
FEES-CONARE: 736-B1-660.

### Competing Interests
The authors declare there are no competing interests.

### Author Contributions
- Marcela Vásquez-Mayorga conceived and designed the experiments, performed the experiments, analyzed the data, contributed reagents/materials/analysis tools, wrote the paper, prepared figures and/or tables, reviewed drafts of the paper.
- Eric J. Fuchs analyzed the data, contributed reagents/materials/analysis tools, wrote the paper, prepared figures and/or tables, reviewed drafts of the paper.
- Eduardo J. Hernández and Natalia M. Barboza conceived and designed the experiments, analyzed the data, contributed reagents/materials/analysis tools, wrote the paper, prepared figures and/or tables, reviewed drafts of the paper.
- Franklin Herrera, Jesús Hernández, Ileana Moreira and Elizabeth Arnáez wrote the paper, reviewed drafts of the paper.

### DNA Deposition
The following information was supplied regarding the deposition of DNA sequences:
KU561375–KU561391, KU561393–KU561424.

## Data Availability

The raw data has been supplied as a Supplementary File.

## Supplemental Information

Supplemental information for this article can be found online at http://dx.doi.org/10.7717/peerj.2931#supplemental-information.

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
