# Peer review of "Molecular characterization and genetic diversity of Jatropha curcas L. in Costa Rica"

_PeerJ, doi:10.7717/peerj.2931_

## Round 0.1 · original submission · Minor Revisions

Both the reviewers are positive about your manuscript. However, they raise several points that needs to be addressed before the manuscript is acceptable for publication. The manuscript definitely needs extensive English edition. I suggest the manuscript be edited by a native English speaker. We would like you to submit a revised manuscript, addressing all the points raised by the reviewers.

Reviewer 1 ·

Basic reporting

The manuscript is followed the PeerJ policies and the templates with data made available. The abstract summarized the overall objective and the outcome of the study. The introduction is concise and gives enough background to the study with prior literature cited extensively. Figures and tables are presented appropriately for the clarity of the result. Relevant raw data were also made available.

Experimental design

The material and methodology followed is sufficiently described for the study to answer the research question of diversity analysis. Rigorous investigation and data analysis has been conducted.

Validity of the findings

The research question is sufficiently answered with the statistically sound data analysis and result presentation. Appropriate conclusions have been drawn and relevant speculations were made.

Additional comments

The manuscript is well planned and executed. The result well presented and discussed. However, extensive English language edition is needed. For instance I picked few examples as below.
L50: period before citation needs to be removed
L58: the word previous is confusing the statement
L59: over the last decade? Or past decade?
L80: Latin America?

·

Basic reporting

The manuscript “Molecular characterization and genetic diversity of Jatropha curcas L. in Costa Rica” by Vasquez-Mayorga investigated the genetic diversity, phylogenetic and toxicity analysis of Jatropha curcas, majority collected from Costa Rica area. They found considerable genetic diversity as measured by expected heterozygosity (HE) and Polymorphism information content (PIC). NJ and ML tree analysis further revealed that accession clustered independently of geographical origin. They also reported one Mexican accession to carry both alleles for toxicity as revealed by PCR based assay. Figure and tables are relevant but need some reorganizing and improvement for readability.

Experimental design

Experiment design and analysis were properly conducted and well explained. One section “SSR analysis” needs more information.

Validity of the findings

Results are well analyzed and support authors conclusion. Authors still need to be careful and not to over interpret about clustering of accessions.

Additional comments

1. Provide information about the toxicity in Jatropha curcas in Introduction about what is known and why is this important.
2. Materials & Methods section, SSR analysis (line 113-120). Please provide the information about scoring of SSR markers, such as presence/absence and/or any size difference of PCR amplicon.
3. Authors emphasize the clustering of germplasm collection independent of geographical origin. This could be artifact of unequal sampling. They used 39 samples from Costa Rica, only one sample from Honduras, Brazil, Uganda, Ecuador, Columbia, El Salvador, South Africa and two from India and Mexico. Topology of trees could have been different if equal number of accession from each geographical region were taken. Author should be careful not to over interpret the result and should write in conservative style.
4. Line 32 “but Costa Rican --- consistently clustered as a separate group” is not consistent with what authors are reporting in main text.
5. Toxicity primers, Table 3 showed 4bp to 20 bp difference in toxic and non-toxic allele and they used agarose 2% gel to resolve the PCR band. To my understanding it is not possible to distinguish such minute difference unless with polyacrylamide gel.
6. Convert table 1 to map and shows geographical regions of different germplasm collection
7. Provide oligo sequences of SCAR primers in table 3
8. First few rows in table 4 are not properly aligned
9. Provide some legend to make it standalone and more legible. Figure 2 is difficult to view. Make same color code as in figure 1.
10. Change the orientation of figure 3 and increase the font size.
11. Typo error, “Fonding” in line 309 and “were” line 338. I guess these should be “Funding” and “where”.

---

## Round 0.2 · accepted · Accept

Authors have addressed all the concerns raised by both the reviewers. The manuscript is now suitable for publication in PeerJ.

·

Basic reporting

Revised manuscript meets standards of PeerJ. Authors carefully addressed the comments/suggestions by reviewers and revised manuscripts accordingly.

Experimental design

Revised appropriately.

Validity of the findings

Revised version meets the standard of PeerJ

Additional comments

Authors carefully addressed the comments of reviewers and revised manuscript accordingly.